# High SLC2A1 expression associated with suppressing CD8 T cells and B cells promoted cancer survival in gastric cancer

**Kyueng-Whan Min**[1©], **Dong-Hoon Kim**[2©], **Byoung Kwan Son**[3*], **Kyoung Min Moon**[4], **So Myoung Kim**[5], **Md. Intazur Rahaman**[5], **So Won Kim**[5], **Eun-Kyung Kim**[6], **Mi Jung Kwon**[7], **Young Wha Koh**[8], **Il Hwan Oh**[3]

1 Department of Pathology, Hanyang University Guri Hospital, Hanyang University College of Medicine, Guri, Gyeonggi-do, Republic of Korea, 2 Department of Pathology, Kangbuk Samsung Hospital, Sungkyunkwan University School of Medicine, Seoul, Republic of Korea, 3 Department of Internal Medicine, Eulji Hospital, Eulji University School of Medicine, Seoul, Republic of Korea, 4 Department of Internal Medicine, Gangneung Asan Hospital, University of Ulsan College of Medicine, Gangneung, Gangwon-do, Republic of Korea, 5 Department of Pharmacology, Asan Medical Center, University of Ulsan College of Medicine, Seoul, Republic of Korea, 6 Department of Pathology, Eulji Hospital, Eulji University School of Medicine, Seoul, Republic of Korea, 7 Department of Pathology, Hallym University Sacred Heart Hospital, Hallym University College of Medicine, Anyang, Gyeonggi-do, Republic of Korea, 8 Department of Pathology, Ajou University School of Medicine, Suwon, Gyeonggi-do, Republic of Korea

© These authors contributed equally to this work.
* sbk1026@eulji.ac.kr

**Data Availability Statement:** All relevant data are within the paper and its Supporting Information files.

## Abstract

High expression of glucose transporter family members, which augment glucose uptake and glycolytic flux, has been shown to play a pivotal role in the proliferation and survival of tumor cells, contributing to the energy supply, biosynthesis and homeostasis of cancer cells. Among the many members, solute carrier family 2 member 1 (SLC2A1) encodes a glucose transporter, GLUT1, that is critical in the metabolism of glucose, which is an energy source for cell growth that contributes to cancer progression and development. The aim of this study was to analyze the survival and genetic changes/immune profiles in patients with gastric cancer with high SLC2A1 expression and to provide treatment for improving prognosis. This study investigated the clinicopathologic parameters, the proportion of immune cells and gene sets affecting SLC2A1 expression in 279 and 415 patients with gastric cancer from the Eulji Hospital cohort and The Cancer Genome Atlas, respectively. We assessed the response to conventional chemotherapy drugs, including fluorouracil, a compound of fluoropyrimidine S-1, oxaliplatin, and all−trans−retinoic acid (ATRA), in gastric cancer cell lines with high SLC2A1 expression. High SLC2A1 expression was associated with poor prognosis, cancer cell proliferation, decreased immune cells, including CD8 T cells and B cells, and a low prognostic nutrition index, representing body nutrition-related status. In pathway network analysis, SLC2A1 was indirectly linked to the retinoic signaling pathway and negatively regulated immune cells/receptors. In the drug response analysis, the drug ATRA inhibited gastric cancer cell lines with high SLC2A1 expression. Treatment involving the use of SLC2A1 could contribute to better clinical management/research for patients with gastric cancer.

**Funding:** YES This study was supported by Daewon Pharmaceutical Co., Ltd. in 2018. (To Byoung Kwan Son). The funding organization did not play any role in our study and provided only financial support. We have reviewed the author roles in the Author Contribution section and confirm the previous statements are correctly stated.

**Competing interests:** This study was supported by Daewon Pharmaceutical Co., Ltd. in 2018. (To Byoung Kwan Son). This does not alter our adherence to PLOS ONE policies on sharing data and materials.

## Introduction

Cancer depends on glycolysis as well as oxidative phosphorylation for energy production, which can affect the proliferation and growth of tumor cells. Glucose metabolism supports cellular homeostasis and maintenance, including transcription, enzymatic activity, hormone secretion, and glucoregulatory neuron activity. Glucose transporter family members (GLUT1-14) are expressed in the membranes of nearly all cell types [1, 2]. Among the many elements related to glucose metabolism, solute carrier family 2 member 1 (SLC2A1), a glucose transporter-encoding gene that controls glucose uptake, could play a pivotal role in the growth and proliferation of tumor cells [3, 4]. In a study by Warburg, tumor cells were seen to take up glucose at an elevated rate to meet their increased energy demands [5]. Glucose transporters facilitate glucose uptake across the plasma membrane and can be enhanced by oncogenes and growth factors [6]. High expression of GLUT1, encoded by SLC2A1, is associated with different types of malignancies, especially those driven by oncogenic KRAS and BRAF mutations or loss of p53, and thereby contributes to the increased proliferation of cancer cells [7–9]. Previous studies demonstrated that high SLC2A1 expression was associated with worse prognosis in colon, lung, breast, and oral cancer [4, 9–11].

Published data have reported that GLUT family proteins affect various aspects of tumor growth and microenvironment components. A study by Macintyre et al. showed a specific requirement for GLUT1 in both activated mouse and human T cells in vitro and in vivo [12]. The study demonstrated that GLUT1 is essential for rapid metabolic reprogramming to aerobic glycolysis for maximal growth, survival, and proliferation of in vitro stimulated T cell functions, especially CD4 T cell differentiation into effector cells. CD8 T cells had reduced initial proliferation in a GLUT1-deficient mouse model, but the levels of granzyme B, interleukin-2, tumor necrotic factor-α and interferon-γ, which are related to the effector function of T cells, were normal [12]. Other studies on colon cancer and diabetes have revealed a significant correlation between impaired expression of GLUT family proteins and decreased activity of natural killer cells, suggesting that GLUT also affects immune system function [13]. Nevertheless, the signaling and pathobiological processes regulated by GLUT1, the major protein of the GLUT family, remain poorly understood in the context of gastric cancer.

In recent years, next-generation sequencing (NGS) and big data analytics have allowed for the analysis of marker genes, the quantification of the different types of tumor-infiltrating immune cells and the molecular network-based integration of multiomics data. Considering the complex gene-environment interactions of gastric cancer, the clinical application of gene expression data is not easy. Analysis using gene expression data should focus on identifying a simple, robust, and druggable biomarker based on bioinformatics and high-throughput experimental methods for accessible and effective therapeutic strategies. According to The Cancer Genome Atlas (TCGA) database, gastric cancer is classified into four molecular subtypes, each with different clinical outcomes and therapeutic strategies [14, 15].

The present study aimed to assess whether SLC2A1 is related to the clinicopathological parameters and survival of patients with gastric cancer in our Eulji Hospital cohort (EHC) and those from the TCGA database [16]. We focused on evaluating SLC2A1-associated immune gene sets and genes, different types of tumor-infiltrating immune cells and network-based pathways as well as in vitro drug screening tests in gastric cancer cell lines.

## Materials and methods

### Patient selection

This study included 279 patients with gastric cancer who underwent surgery at Eulji Hospital in Korea between 2004 and 2014. The Reporting Recommendations for Tumor Marker

Prognostic Studies (REMARK) criteria were followed throughout this study [17]. The inclusion criteria were as follows: 1) patients with microscopic features of primary gastric adenocarcinoma confirmed by pathologists and with known medical records; and 2) patients who did not undergo concurrent neoadjuvant chemoradiotherapy. Cases with missing paraffin blocks of tumor samples or incomplete clinical outcomes were excluded. We assessed T and N stage, location, size, Lauren type, [18] histopathological grade/differentiation, lymphovascular and perineural invasion, recurrence/metastasis and Epstein-Barr virus (EBV) status (**S1 Table**). Before cancer treatment, the prognostic nutrition index (PNI) was calculated as $10 \times$ serum albumin (g/dL) + $0.005 \times$ total lymphocyte count (/mm$^3$) [19].

**Ethics approval.** This study (involving human participants) was approved by the Ethics Committee of the Eulji Hospital, Seoul, Republic of Korea (EMCIRB 2018-09-01), and was performed according to the ethical standards of the Declaration of Helsinki, as revised in 2008. The need of informed consent was waived by institutional review board (Eulji medical center institutional review board who reviewed the study. The patients' medical records and samples were fully anonymized before we accessed them in September 2018.

## Cell line management

MKN-45 cells (KCBL 80103, Korean Cell Line Bank, Korea) were maintained in RPMI 1640 (LM011-03, Welgene, Korea) supplemented with 10% FBS (16000044, Gibco, USA). Cells were incubated at 37°C in a 5% CO2 humid incubator (Heracell VIOS 160i, 51030287, Thermo Fisher, USA) (**S1 File**)

## Tissue microarray construction and immunohistochemistry

The tissue microarray (TMA) blocks were assembled using a tissue array instrument (Accu-Max Array; ISU ABXIS Co., Ltd., Seoul, Korea). We used duplicate 3-mm-diameter tissue cores (tumor component in a tissue core > 70%) from each donor block. Four-micrometer sections were cut from the TMA blocks using routine techniques. Immunostaining for SLC2A1 (1:200; Cell Marque, Rocklin, CA, USA) was performed using the Dako Autostainer Universal Staining System (DakoCytomation, Carpinteria, CA, USA) and the ChemMate™ Dako EnVision™ Detection Kit. SLC2A1 expression was graded according to the intensity and the proportion of membranous-stained tumor cells [20] (**Fig 1A**). The immunoreactive score (IRS) was calculated (intensity × proportion), and SLC2A1 expression was determined as either low (IRS < 1) or high (IRS ≥ 1) using a receiver operating characteristic (ROC) curve. In addition, immunostaining for human epidermal growth factor receptor 2 (HER2) (1:200; Ventana Medical Systems, Tucson, AZ, USA), programmed death-ligand 1 (PD-L1) (clone SP142, Ventana Medical Systems, Tucson, AZ, USA), anti-CD8 (clone 4B11 Leica Biosystems, Newcastle, UK) and anti-CD4 (clone 4B12 Leica Biosystems, Newcastle, UK) was performed. According to the College of American Pathologist (CAP), HER2 was defined as positive in samples with membranous reactivity in ≥10% of tumor cells [21]. According to the tumor proportion score, PD-L1 positivity was defined based on the percentage of tumor cells that stained positive (membranous reactivity) [22]. In situ hybridization (ISH) detection of EBV using probes directed against Epstein-Barr virus-encoded RNA was performed using an EBV ISH kit (Leica Biosystem, Newcastle Ltd., Newcastle, UK).

Twelve-millimeter Φ cover glasses were placed into 24-well plates and incubated with poly-D-lysine hydrobromide (P6407, Sigma) at room temperature for 10 min. Cover glasses were washed with distilled water and dried in air. MKN-45 cells were seeded at $1 \times 10^5$ cells per well. After 24 hours, cells were exposed to DMSO or 1 μM retinoic acid (ATRA, all-trans-retinoic acid, R2625, Sigma) at 37°C and 5% CO2 for 24 hours. Cells were washed with cold PBS

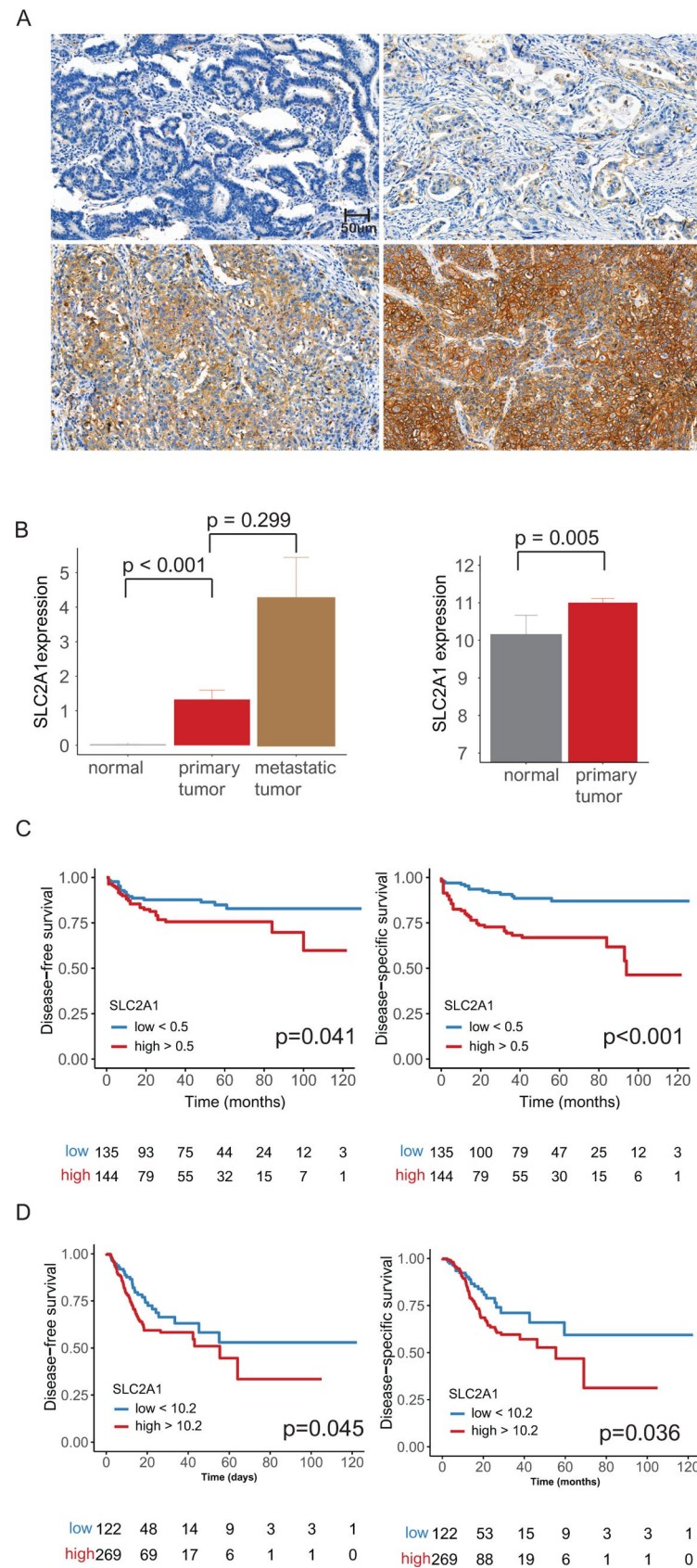

**Fig 1.** (**A**) Representative microphotographs showing negative (top left), weak (top right), moderate (bottom left) and strong intensity (bottom right) SLC2A1 expression in gastric adenocarcinoma by immunohistochemical staining (original magnification x400). (**B**) Bar plots of SLC2A1, Eulji Hospital cohort paired (matched) samples: SLC2A1 expression was highest in metastatic tumors followed by primary tumors (left). TCGA paired (matched) samples: high SLC2A1 expression was seen in primary tumors compared to that in normal tissue samples (right) (error bars: standard errors of the mean). (**C**) Eulji Hospital cohort: high SLC2A1 expression was associated with poor disease-free and disease-specific survival in 279 patients ($p = 0.039$ and $0.001$, respectively). (**D**) TCGA data: high SLC2A1 expression was associated with poor disease-free and disease-specific survival in 415 patients ($p = 0.045$ and $0.036$, respectively).

(IBS-BP007, iNtRON) for 5 min. Cells were fixed using 4% paraformaldehyde (P2031, Biosesang, Korea) at 4°C for 10 min and washed with PBS 3 times for 10 min. The blocking process was carried out at 37°C for 1 hour. Cells were incubated at 4°C overnight with anti-glucose transporter GLUT1 antibody (ab40084, Abcam), which was diluted at 5 μg/ml in blocking solution. Then, the cells were washed with PBS 3 times for 10 min and incubated with 1 μg/ml cross-adsorbed secondary antibody, Alexa Fluor 488 (A-11001, Invitrogen), diluted in blocking solution at room temperature for 1 hour. Cells were washed with PBS 3 times and washed with 1% PBST. The mounting process was carried out using Fluoroshield Mounting Medium with DAPI (ab104139, Abcam).

## Measurement of the cell growth rate

MKN-45 cells were seeded at 3,400 cells per well in 96 well plate. The experiments were performed in triplicate for each concentration, and each experiment was conducted three times independently. Twenty-four hours after cell seeding, cells were exposed to 0.01, 0.1, 1, 10, or 100 μM retinoic acid. At 0 and 72 hours after retinoic acid treatment, a WST-8 cell viability assay was performed using a Quantimax Cell Viability Assay Kit (QM2500, BIOMAX, Korea). A 10% concentration per well was applied to load the agent to a total of 100 μl/well and incubated at 37°C in a 5% CO2 incubator for 1 hour. The absorbance was then measured using a microplate spectrophotometer (Epoch, Biotek, USA) at 450 nm and 600 nm. The formazan produced by the reaction of tetrazolium salt with dehydrogenase was measured at 450 nm, and the turbidity was measured at 600 nm to subtract the OD-600 from the OD-450 value. OD-0h was subtracted from OD-72h for each concentration to calculate the growth rate of the cells.

## Gene set enrichment analysis, in silico cytometry, and network analyses

We obtained 415 gastric cancer cases with corresponding mRNA expression, mutation, copy number variation, and methylation data from the TCGA database (https://portal.gdc.cancer.gov/) [23]. We analyzed significant gene sets using gene set enrichment analysis (GSEA, version 4.3) from the Broad Institute at MIT [24]. The immunologic gene sets (4,872 sets) were used to identify the gene sets associated with high SLC2A1 expression. For this analysis, 1,000 permutations were used to calculate the $p$-values, and the permutation type was set to phenotype; the following cut-offs were used: $p < 0.05$ and false discovery rate (FDR) of $< 0.4$.

We applied CIBERSORT, also known as in silico cytometry, to determine the proportions of 22 subsets of immune cells using 547 genes [25]. Gene expression datasets were prepared using standard annotation files, the data were uploaded to the CIBERSORT web portal, and the algorithm was run using the default signature matrix at 1,000 permutations [25].

The pathway network analyses were visualized using Cytoscape (version 3.7.2) software. To interpret the biological relevance of SLC2A1 and its relevant elements in gastric cancer, we performed functional enrichment analysis to clarify functionally grouped gene ontology and pathway annotation networks using ClueGO (version 2.5.5) [26, 27].

## Data extraction from the GDSC database

We analyzed the relationship between anticancer drug sensitivity and SLC2A1 expression based on the Genomics of Drug Sensitivity in Cancer (GDSC) dataset (https://www.cancerrxgene.org/celllines) [28]. Eight gastric cancer cell lines were divided into high and low groups based on the median value of SLC2A1 expression. In gastric cancer cell lines with low SLC2A1 expression (cell lines: IM-95, GCIY, TGBC11TKB, and SK-GT-2; SLC2A1 < 0 based on the z-score) or high SLC2A1 expression (cell lines: MKN45, NUGC-3, RERF-GC-1B, and KATOIII; SLC2A1 > 0), the drug response was defined as the natural log of the half-maximal inhibitory concentration (LN IC50). A drug was identified as an effective SLC2A1-targeting drug when the calculated LN IC50 value was decreased in cell lines with high SLC2A1 expression and increased in those with low SLC2A1 expression, i.e., when an inverse correlation was observed. Pearson's correlation analysis between the LN IC50 values and SLC2A1 expression was also performed [29, 30].

## Statistical analysis

Correlations between clinicopathological parameters and SLC2A1 were analyzed using the $\chi^2$ test and a linear-by-linear association test. Student's t-test and/or Pearson's correlation analysis were used to examine the differences among continuous variables. Disease-free survival (DFS) was defined as the time from the date of diagnosis to recurrence/new distant metastasis, with disease-specific survival (DSS) defined as the time from the date of diagnosis to cancer-related death. Survival curves were generated using the Kaplan–Meier method and then compared using the log-rank test. Multivariate Cox regression analyses were performed to identify independent prognostic markers for DFS and DSS. A two-tailed $p$-value of $< 0.05$ was considered statistically significant. All data were analyzed using R packages and SPSS statistics (version 25.0, SPSS Inc., Chicago, IL, USA).

## Results

### Clinical manifestations of SLC2A1

In the EHC, SLC2A1 expression was evaluated in 189 normal mucosa, 279 primary cancer and 58 metastatic cancer samples. We analyzed SLC2A1 expression among normal and primary tumor or metastatic tumor paired (matched) samples. We have analyzed 189 normal and 58 metastatic tumor samples from 279 primary cancer samples. Compared to that in normal mucosa, SLC2A1 expression was significantly higher in primary cancers (189 normal mucosa versus 189 primary tumor samples, $p < 0.001$). On the other hand, SLC2A1 expression was higher in metastatic cancers than in primary cancers (58 primary tumor versus 58 metastatic tumor samples, $p = 0.299$) (**Fig 1B, left**). In the TCGA data (survival data: 391 cases), primary cancer tissues showed higher SLC2A1 expression than normal tissues ($p = 0.005$) (**Fig 1B, right**).

In the EHC, high SLC2A1 expression was significantly associated with advanced T stage, advanced N stage, large tumor size, diffuse type, high histological grade, lymphatic invasion, high PD-L1 expression, low PNI, and chemoresistance, compared with low SLC2A1 expression ($p = 0.001, 0.001, 0.003, 0.002, 0.001, 0.001, 0.028, 0.048$ and $0.002$, respectively) (**Tables 1 and S1**). High SLC2A1 expression was significantly correlated with worse DFS and DSS compared to low SLC2A1 expression ($p = 0.041$ and $< 0.001$, respectively) (**Table 2**) (**Fig 1C**). In multivariate analyses, there was still a significant relationship between SLC2A1 and DSS ($p = 0.005$). In the TCGA data, high SLC2A1 expression was significantly associated with poor DFS and DSS ($p = 0.045$ and $0.036$, respectively) (**Fig 1D**).

**Table 1. Correlation between clinicopathological parameters and SLC2A1 expression in 279 gastric cancer patients (Eulji Hospital cohort).**

| Parameters | N = 279 | SLC2A1 expression | | P-value |
| --- | --- | --- | --- | --- |
| | | Low (n = 135), % | High (n = 144), % | |
| Age (year) | | | | |
| <65 | 84 | 45 (33.3) | 39 (27.1) | 0.255[1] |
| ≥65 | 195 | 90 (66.7) | 105 (72.9) | |
| Sex | | | | |
| Male | 179 | 87 (64.4) | 92 (63.9) | 0.923[1] |
| Female | 100 | 48 (35.6) | 52(36.1) | |
| T stage | | | | |
| 1 | 158 | 92 (68.1) | 66 (45.8) | **<0.001**[2] |
| 2 | 21 | 10 (7.4) | 11 (7.6) | |
| 3 | 52 | 18 (13.3) | 34 (23.6) | |
| 4 | 48 | 15 (11.1) | 33 (22.9) | |
| N stage | | | | |
| 0 | 177 | 101 (74.8) | 76 (52.8) | **<0.001**[2] |
| 1 | 25 | 12 (8.9) | 13 (9.0) | |
| 2 | 21 | 4 (3.0) | 17 (11.8) | |
| 3 | 56 | 18 (13.3) | 38 (26.4) | |
| Location | | | | |
| Cardia, fundus body | 102 | 55 (40.7) | 47 (32.6) | 0.16[1] |
| Antrum or pylorus | 177 | 80 (59.3) | 97 (67.4) | |
| Size | | | | |
| ≤ 3 cm | 123 | 72 (53.3) | 51 (35.4) | **0.003**[1] |
| > 3 cm | 156 | 63 (46.7) | 93 (64.6) | |
| Lauren type | | | | |
| Intestinal | 175 | 72 (53.3) | 103 (71.5) | **0.002**[3] |
| Diffuse | 61 | 46 (34.1) | 15 (10.4) | |
| Mixed | 43 | 17 (12.6) | 26 (18.1) | |
| Histological grade | | | | |
| Well differentiated | 41 | 24 (17.8) | 17 (11.8) | **<0.001**[4] |
| Moderately differentiated | 117 | 37 (27.4) | 80 (55.6) | |
| Poorly differentiated | 54 | 26 (19.3) | 28 (19.4) | |
| Signet ring[5] | 67 | 48 (35.6) | 19 (13.2) | |
| Lymphatic invasion | | | | |
| Not identified | 155 | 92 (68.1) | 63 (43.8) | **<0.001**[1] |
| Present | 124 | 43 (31.9) | 81 (56.2) | |
| Vascular invasion | | | | |
| Not identified | 233 | 127 (94.1) | 106 (73.6) | **<0.001**[1] |
| Present | 46 | 8 (5.9) | 38 (26.4) | |
| Perineural invasion | | | | |
| Not identified | 223 | 113 (83.7) | 110 (76.4) | 0.127[1] |
| Present | 56 | 22 (16.3) | 34 (23.6) | |
| Epstein-Barr Virus | | | | |
| Absent | 242 | 121 (89.6) | 121 (84.0) | 0.168[1] |
| Present | 37 | 14 (10.4) | 23 (16.0) | |
| HER2 | | | | |
| Negative | 257 | 125 (98.4) | 132 (95.7) | 0.285[5] |
| Positive | 8 | 2 (1.6) | 6 (4.3) | |

(*Continued*)

**Table 1.** (Continued)

| Parameters | N = 279 | SLC2A1 expression | | P-value |
|---|---|---|---|---|
| | | Low (n = 135), % | High (n = 144), % | |
| PD-L1 | | | | |
| Negative | 189 | 100 (74.1) | 89 (61.8) | **0.028**[1] |
| Positive | 90 | 35 (25.9) | 55 (38.2) | |
| Prognostic nutritional index | | 52.56 ± 1.4 | 49.13 ± 1.04 | **0.048**[6] |
| Adjuvant chemotherapy[7] | | | | |
| Sensitive | 121 | 67 (90.5) | 54 (70.1) | **0.002** |
| Resistant | 30 | 7 (9.5) | 23 (29.9) | |

HER2, human epidermal growth factor receptor 2; PD-L1, programmed death-ligand 1.

[1] $\chi^2$ test.

[2] linear-by-linear association test.

[3] intestinal type versus diffuse or mixed type.

[4] well or moderately differentiated type versus poorly differentiated or signet ring type.

[5] Fisher's exact test.

[6] Student's t-test.

[7] One hundred fifty-one patients with postoperative adjuvant chemotherapy.

$p < 0.05$ is shown in bold.

## SLC2A1 expression in relation to mutation, copy number alteration, and methylation status

In analyses of SLC2A1 mutations, SLC2A1 expression was elevated in mutant compared to wild-type samples ($p = 0.098$). SLC2A1 expression was increased in samples with copy number gain/high-level amplification compared with that in samples with neutral changes/no change in copy number ($p = 0.047$). In analyses of methylation using the Human Methylation 450K

**Table 2. Disease-free survival and disease-specific survival according to SLC2A1 expression in 279 patients with gastric cancer (Eulji Hospital cohort).**

| Disease-free survival | Univariate[1] | Multivariate[2] | HR | 95% CI | |
|---|---|---|---|---|---|
| SLC2A1 (low vs. high) | **0.041** | 0.283 | 0.689 | 0.350 | 1.359 |
| T stage (1 or 2 vs. 3 or 4) | **<0.001** | 0.088 | 2.167 | 0.892 | 5.268 |
| N stage (0 vs. 1, 2 or 3) | **<0.001** | **<0.001** | 6.950 | 2.707 | 17.843 |
| Histological grade (1 or 2 vs. 3) | 0.442 | 0.111 | 0.596 | 0.315 | 1.126 |
| Vascular invasion (absence vs. presence) | **<0.001** | **0.005** | 2.451 | 1.303 | 4.612 |
| Perineural invasion (absence vs. presence) | **<0.001** | 0.087 | 1.756 | 0.922 | 3.345 |
| Disease-specific survival | Univariate[1] | Multivariate[2] | HR | 95% CI | |
| SLC2A1 (low vs. high) | **<0.001** | **0.005** | 2.543 | 1.330 | 4.865 |
| T stage (1 or 2 vs. 3 or 4) | <0.001 | **0.025** | 2.535 | 1.125 | 5.713 |
| N stage (0 vs. 1, 2 or 3) | **<0.001** | **0.02** | 2.438 | 1.151 | 5.163 |
| Histological grade (1 or 2 vs. 3) | 0.145 | 0.536 | 1.194 | 0.682 | 2.089 |
| Vascular invasion (absence vs. presence) | **<0.001** | 0.362 | 1.320 | 0.727 | 2.397 |
| Perineural invasion (absence vs. presence) | **<0.001** | 0.369 | 1.316 | 0.723 | 2.395 |

HR, hazard ratio; CI, confidence interval.

1Log rank test.

2Cox proportional hazard model.

$p < 0.05$ is shown in bold.

platform, the beta (β)-value for the hypermethylation cut-off was defined as 0.2 [31]. In 373 cases with methylation data, low SLC2A1 expression was associated with hypermethylation ($p < 0.001$) (**Fig 2**).

## Gene set enrichment analysis, immune cell proportion and pathway network analysis of SLC2A1

In the TCGA database, we conducted GSEA to identify the genes associated with high SLC2A1 expression. We found four significantly enriched gene sets related to the negative regulation of immune cells (GSE20715: "0 hour vs 48 hour Ozone Toll-like receptor 4 KO Down"; GSE15930: "Naive vs In vitro CD8 T cell Down"; GSE3982: "Memory CD4 T cell vs Th1 cell Down"; and GSE6674: "Anti IgM vs Anti IgG2a Stimulated B cell Down") in immunologic gene sets (**Fig 3A**). On the basis of GSEA, we analyzed the relationships between SLC2A1 and immune-related elements. In the EHC, CD8+ T cells were elevated in patients with high SLC2A1 expression compared to those with low SLC2A1 expression ($p = 0.049$). CD4+ T cells were lower in high SLC2A1 than in low SLC2A1 expression, but this difference was not statistically significant ($p = 0.501$) (**Fig 3B**). In TCGA, high SLC2A1 expression was associated with decreased tumor-infiltrating lymphocytes (TILs), CD8 T cells, B cells, T cell receptor (TCR) expression and B cell receptor (BCR) expression ($p = 0.001, 0.004, 0.009, 0.001$ and $0.009$, respectively) (**Fig 3C**), while its expression was associated with increased proliferation and cancer/testis antigen (CTA) expression ($p = 0.001$ and $0.009$, respectively) (**Fig 3D**). In pathway network analysis based on GSEA, we found that high SLC2A1 expression was indirectly linked to negative regulation of immune cells, BCR signaling, and the retinoic acid pathway (**Fig 3E**).

## Drug screening in gastric cell lines with high SLC2A1 and retinoic acid receptor expression

On the basis of the GDSC data, we analyzed drug sensitivity patterns in 8 gastric cancer cell lines with high SLC2A1 expression based on ATRA, known all−trans−retinoic acid,

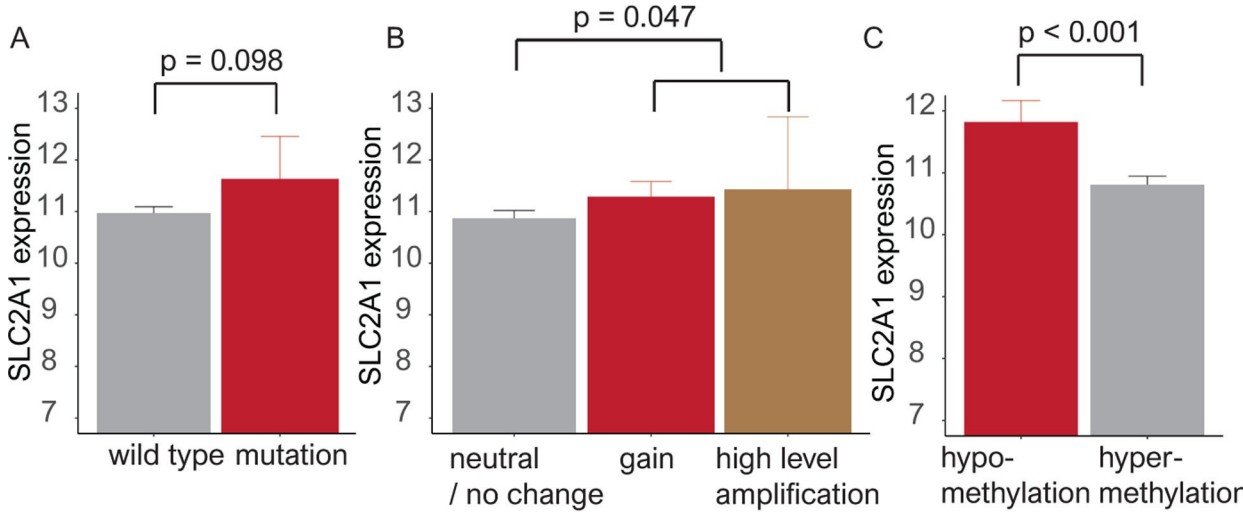

**Fig 2. Bar plots showing SLC2A1 expression according to mutation, copy number alteration and methylation.** (**A**) SLC2A1 expression is elevated in the mutant type compared with the wild type ($p = 0.098$). (**B**) SLC2A1 is highly expressed in copy number gain/amplification compared to neutral/no change ($p = 0.047$). (**C**) Hypermethylation was associated with a decline in SLC2A1 expression ($p < 0.001$).

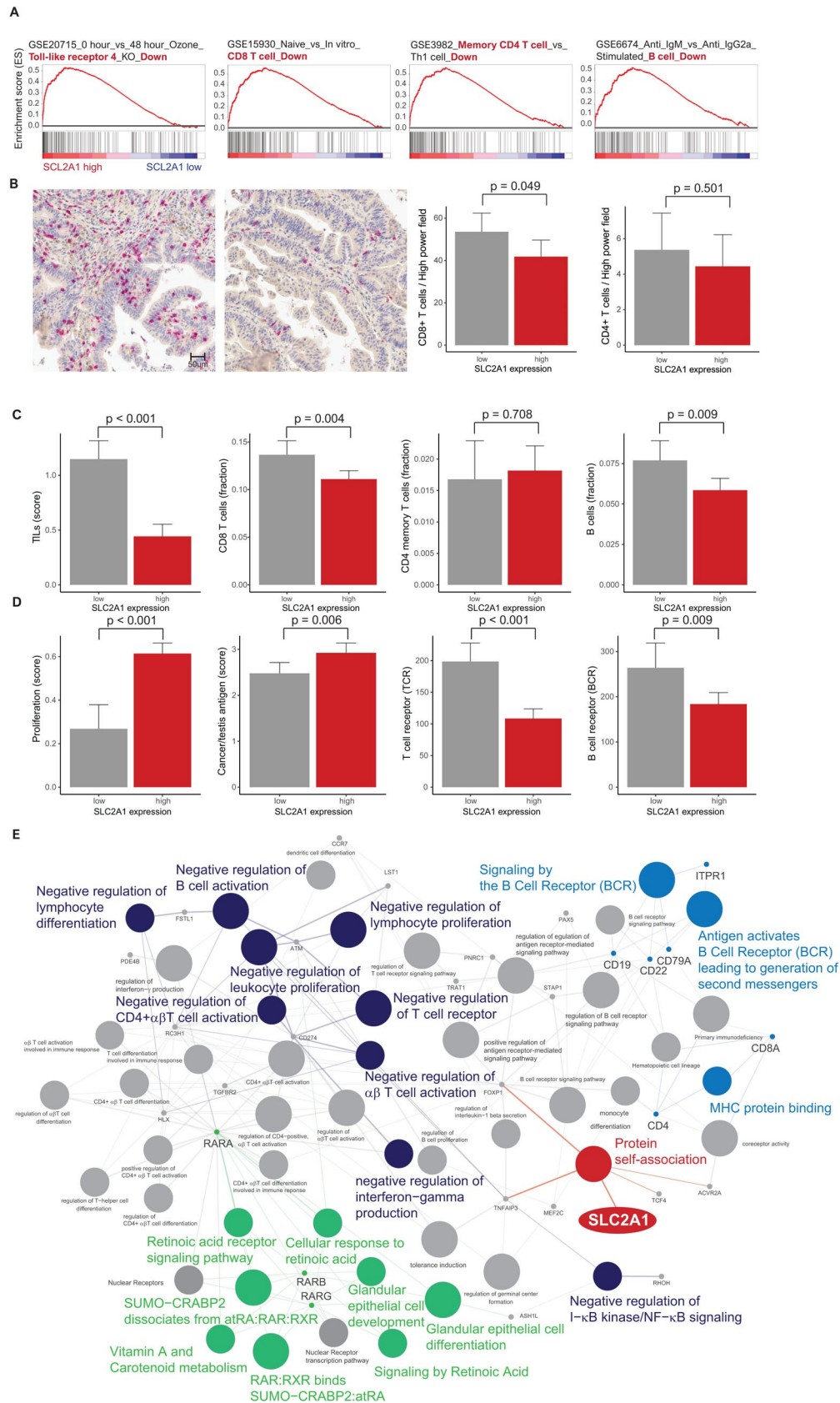

**Fig 3.** (**A**) Gene set enrichment analysis (GSEA) of four SLC2A1-dependent immunologic gene sets: downregulation of toll-like receptor 4 (TLR-4) expression, CD8 T cells, memory CD4 T cells and B cells. (**B**) Representative microphotographs showing CD8 T cells (red): increased CD8 T cells and decreased CD8 T cells in low SLC2A1 expression (left) and high SLC2A1 expression (right), respectively. Bar plot of CD8 T cells (left) and CD4 T cells (right) per high-power field (*p* = 0.049 and 0.501, respectively) in our cohort. (**C**) Bar plot of tumor-infiltrating lymphocytes (TILs), CD8 T cells, CD4 memory T cells and B cells between samples with low (gray) and high (red) SLC2A1 expression (*p* = 0.001, 0.004, 0.708 and 0.009, respectively) in the TCGA database. (**D**) Bar plot of proliferation, cancer/testis antigen (CTA) expression, T cell receptor (TCR) expression and B cell receptor (BCR) expression between samples with low (gray) and high (red) SLC2A1 expression (*p* = 0.001, 0.006, 0.001 and 0.009, respectively) (error bars: standard errors of the mean). (**E**) Grouping of networks based on functionally enriched gene ontology (GO) terms and pathways: SLC2A1 (black) was indirectly associated with negative regulation of immune cells (red), immune receptors (blue) and retinoids (green).

fluorouracil, a compound of oral fluoropyrimidine S-1, and oxaliplatin, known as XELOX [32]. Using Pearson's correlation, we considered drugs exhibiting a high negative correlation between SLC2A1 and the LN IC50 value as effective SLC2A1-targeting drugs. ATRA most effectively reduced the growth of cancer cell lines with high SLC2A1 expression [ATRA: r = -0.727, *p* = 0.041 (Pearson's correlation) and 0.028 (Student's t-test); fluorouracil: r = -0.427, *p* = 0.292 and 0.154; oxaliplatin: r = 0.353, *p* = 0.437 and 0.562] (**Fig 4A and 4B**). In the analysis of the relationships between SLC2A1 and retinoic acid receptors (RARs)/retinoic X receptor (RXRs), including RARα, RARβ, RARγ, RXRα, RXRβ and RXRγ, high SLC2A1 expression was related to low RARβ, RXRα and RXRγ and high RARγ (*p* = 0.011, 0.028, 0.002 and 0.001, respectively) (**Fig 4C and 4D**).

## Determination of the biological effectiveness of retinoic acid in MKN-45 cells

Upon exposure to retinoic acid, SLC2A1 mRNA expression and GLUT1 protein expression were elevated in MKN-45 cells known for their high SLC2A expression from the GDSC dataset. There were also some mRNAs and proteins of SLC2A1 in non-drug-treated control-group cells, but they increased further when exposed to retinoic acid (**Figs 5A and 5B and S1**). In the protein analysis using the immunocytochemical method, the expression of GLUT1 protein increased at the location of the membrane around the DAPI-stained area after retinoic acid treatment (**Fig 5C**).

Retinoic acid at concentrations greater than $10^2$ mM inhibited the growth rate of MKN-45 cells. When the MKN-45 cells were treated with retinoic acid for 72 h at 0.01, 0.1, 1, 10, and 100 μM, the cells showed growth rates of 84, 78, 73.4, 58.6 and -21%, respectively, compared to the control group. The half maximal growth inhibition concentration (GI50) and GI100 values were calculated as 19.8 and 76.6 μM, respectively (**Fig 5D**) [33].

## Discussion

SLC2A1 can enhance intracellular glucose as an energy source and thereby provide favorable conditions for tumor growth and subsequent dissemination and metastasis. This study demonstrated that compared with low SLC2A1, high SLC2A1 was related to worse clinical outcomes, such as advanced T and N stage, large tumor size, lymphatic invasion, mutation, copy number gain/amplification and hypomethylation in patients with gastric cancer. SLC2A1 was more highly expressed in metastatic tumors than in primary tumors. In survival analyses, compared with low SLC2A1 expression, high SLC2A1 expression was associated with worse DFS and DSS in patients with gastric cancer. Interestingly, there was a negative correlation between the PNI and SLC2A1 expression. Moreover, PD-L1, as a marker for determining the use of immunotherapy, was highly expressed in gastric cancer with high SLC2A1 expression.

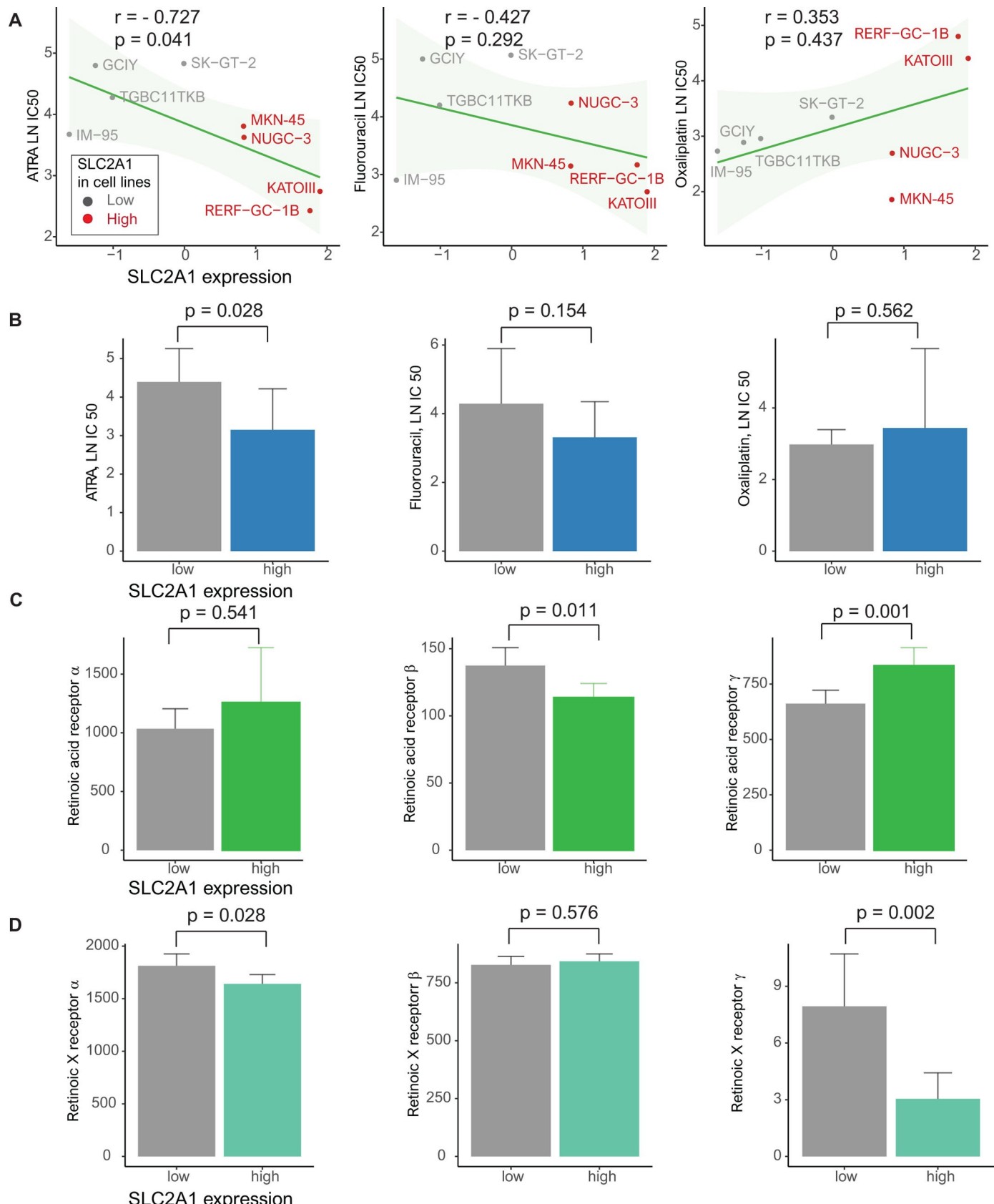

**Fig 4. Genomics of drug sensitivity in cancer (GDSC) database analysis.** (**A**) Pearson's correlations showing the natural log of the half-maximal inhibitory concentration (LN IC50) values of ATRA, fluorouracil and oxaliplatin in gastric cancer cell lines (gray, low SLC2A1 expression; red, high SLC2A1 expression). (**B**) Bar plot showing the LN IC50 values of ATRA, fluorouracil and oxaliplatin between gastric cancer cell lines with low (gray) and high (blue) SLC2A1 expression ($p = 0.022$, 0.852 and 0.377, respectively) (error bars: standard errors of the mean). (**C**) TCGA database: bar plot of the expression of the retinoic acid receptors (RARs) RARα, RARβ and RARγ between patients with gastric cancer with low (gray) and high (green) SLC2A1 expression ($p = 0.541$, 0.011 and 0.001, respectively) (error bars: Standard errors of the mean). (**D**) TCGA database: Bar plot of the expression of the retinoic X receptors (RXRs) RXRα, RXRβ and RXRγ between patients with gastric cancer with low (gray) and high (blue green) SLC2A1 expression ($p = 0.028$, 0.576 and 0.002, respectively) (error bars: Standard errors of the mean).

To further reinforce the implications of these findings, we analyzed the association of SLC2A1 with survival data from TCGA, a large-scale database, to improve the reproducibility of the findings. As seen previously in our study, high SLC2A1 expression was related to poor DFS and DSS. Thus, we suggest that SLC2A1 could play an important role in promoting cancer progression.

Several studies have demonstrated that SLC2A1 overexpression is associated with poor clinical outcomes in various types of malignancies [4, 9, 10, 34], but the precise mechanisms by which SLC2A1 could elevate glucose uptake in cancer cells are not fully understood. One hypothesis is that SLC2A1 increases glucose metabolism and provides a high energy source for cancer cells. A recent study of lung cancer demonstrated that high SLC2A1 expression was associated with increased glucose uptake on PET-CT [10]. In our results, an inverse

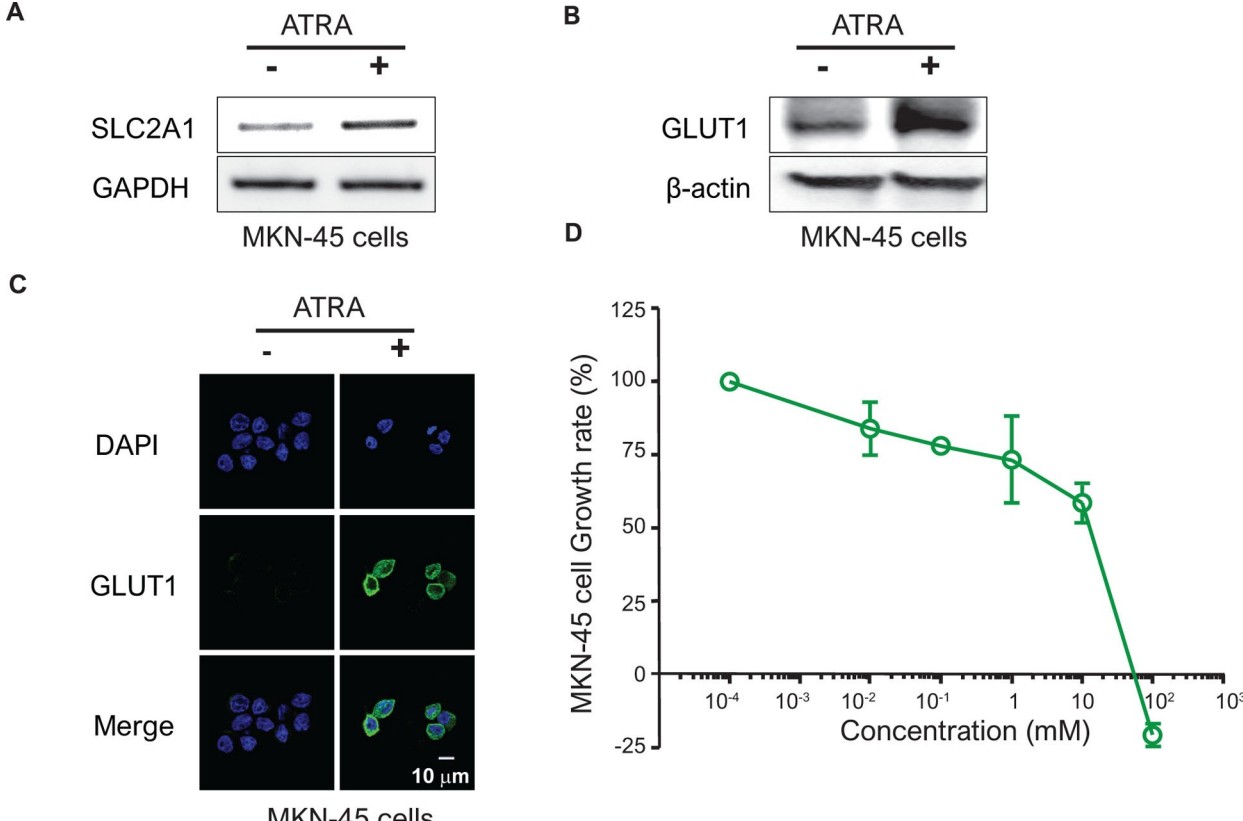

**Fig 5. Determination of the biological efficacy of retinoic acid in MKN-45 cells.** After treating MKN cells with 1 μM retinoic acid for 24 hours, SLC2A1 mRNA and GLUT1 protein expression was checked against the control group by (A) RT-PCR and (B) immunoblotting, respectively. GAPDH and β-actin were used as internal controls. (C) Under the same conditions as (A), the nucleus (DAPI) and GLUT1 expression and location were identified by immunocytochemistry. (D) The survival rate of MKN-45 cells was observed after 72 hours of treatment with retinoic acid in MKN-45 cells at concentrations ranging from 10 nM to 100 μM.

relationship between SLC2A1 in tumor cells and the PNI indirectly showed that glucose could not be transferred to normal cells for the maintenance of energy homeostasis or concentrated during cancer progression.

In computational analyses such as GSEA, in silico cytometry and pathway network analyses, our results revealed that SLC2A1-related gene sets were associated with negative regulation of immune cells and components such as Toll-like receptors, CD8 T cells, CD4 T cells and B cells. High SLC2A1 expression was related to decreased TILs, CD8 T cells, B cells, TCR signaling and BCR signaling, whereas it was related to increased proliferation and cancer/testis antigen expression. This suggests that SLC2A1 may affect immune cells, as well as cancer growth suppression. The negative association between SLC2A1 and immune cells may be important for designing immunotherapies for the treatment of gastric cancer. In pathway network analysis, the SLC2A1 pathway was indirectly linked to the RAR signaling pathway as well as glandular epithelial development. Further experimental studies are necessary to prove these relationships among the various factors associated with SLC2A1.

The GDSC database, which contains data from pharmacogenomic screens in cancer cell lines, uses an unbiased discovery approach for putative markers of drug sensitivity [30]. Given the link between SLC2A1 and retinoic acid, we investigated the sensitivity to ATRA between gastric cancer cell lines with high SLC2A1 expression and those with low SLC2A1 expression. ATRA was effective in gastric cancer cell lines exhibiting high SLC2A1 expression. An RXR selective ligand, bexarotene, was not effective in gastric cancer cell lines exhibiting high SLC2A1 expression (data not shown). A previous study demonstrated that retinoic acid could enhance antigen presentation in retinoid-treated dendritic cells, which activate T cells [35]. In our study, RARγ was increased in cells with high SLC2A1 expression, but RARβ, RXRα and RXRγ were decreased in cells with high SLC2A1 expression. There was a difference in expression according to RAR/RXR subtypes. Another study of retinoic acid reported that increased RARα and RARγ could mediate growth inhibition by all-trans retinoic acid (ATRA) in H1792 cells, a lung adenocarcinoma cell line [36]. However, interactive molecules and pathways of targeted drugs for gastric cancer with high SLC2A1 expression have not yet been elucidated.

We analyzed the sensitivity to fluorouracil, a compound of oral fluoropyrimidine S-1, and oxaliplatin, which are adjuvant chemotherapies for patients with gastric cancer, in gastric cancer cell lines [32]. Gastric cancer cell lines with high SLC2A1 expression were more sensitive to oxaliplatin than those with low SLC2A1 expression, but the difference was not statistically significant. An in vitro study to evaluate the inhibitory effect of ATRA in gastric cancer cell lines with high SLC2A1 expression revealed that a high concentration (over $10^2$ mM) of retinoic acid significantly suppressed the growth of MKN-45 cells with high SLC2A1 expression [37]. ATRA, known as retinoic acid, inhibited gastric cancer cells with high SLC2A1 expression in this study, but there are some considerations for the clinical application of this drug. Unlike the responses in cell lines with high SLC2A1 expression, the therapeutic responses in patients with gastric cancer may be highly heterogeneous and affected by various microenvironments and immune components, which could have effects on clinical applications. Furthermore, some cell lines may be partially sensitive or resistant to a given drug within the range of experimental screening concentrations. Therefore, interpretations based on LN IC50 values could have limited utility in explaining drug sensitivity. Along with in vivo studies, ATRA-based clinical trials in gastric cancer with high SLC2A1 expression are needed in the future.

This study had some limitations that should be acknowledged. First, because this is a retrospective study and because the analyses of SLC2A1 did not show sustained relationships over time as prospective studies do, it is difficult to come to a definitive conclusion. Second, experimental results allowing for novel biological insights into the relationship between SLC2A1 and

immune cells were not shown, and further in vivo studies may be necessary. Third, our study did not investigate the relationship between SLC2A1 expression and glucose uptake based on PET-CT results in cancer. Further studies are necessary to prove the relationship between SLC2A1 and glucose in gastric cancer cells.

In summary, the study demonstrated that high SLC2A1 expression was statistically associated with poor DFS/DSS as well as copy number gain/amplification and hypomethylation in patients with gastric cancer in both our EHC and TCGA databases. In gastric cancer with high SLC2A1 expression, the decrease in immune cells and immune components, such as CD8 T cells and B cells and TCR, BCR and PD-L1 expression, is related to type III (intrinsic) induction. Without TILs in the tumor, it is unlikely that blocking PD-L1 will lead to a T cell response to cancer [38]. As an alternative to immunotherapy, ATRA could be a candidate drug for the treatment of patients with high SLC2A1 expression and resistance to conventional chemotherapy.

We believe that medical oncologists and researchers will be interested in the role of SLC2A1 in contributing to the energy supply for the development and growth of gastric cancer and that our results will facilitate further studies. In addition, our analytic workflow for SLC2A1 will contribute to designing future experimental studies and future drug development for patients with gastric cancer.

## Supporting information

**S1 File. Reverse transcription polymerase chain reaction (RT-PCR).**
(PDF)

**S1 Fig. Full-length gels and blots.** The GLUT1 protein has a size of 55 kDa but varies slightly in shape depending on cell, antibody or experimental conditions.
(PDF)

**S1 Table. Clinicopathological parameters of Eulji cohort.**
(PDF)

## Author Contributions

**Conceptualization:** Kyueng-Whan Min, Dong-Hoon Kim, Byoung Kwan Son, Young Wha Koh.

**Data curation:** Kyueng-Whan Min, Dong-Hoon Kim, Byoung Kwan Son.

**Formal analysis:** Kyueng-Whan Min, Kyoung Min Moon, Md. Intazur Rahaman, So Won Kim, Mi Jung Kwon.

**Funding acquisition:** Byoung Kwan Son.

**Investigation:** Kyoung Min Moon, So Myoung Kim, Md. Intazur Rahaman, So Won Kim, Mi Jung Kwon, Il Hwan Oh.

**Methodology:** Kyueng-Whan Min, Kyoung Min Moon, So Myoung Kim, Md. Intazur Rahaman, So Won Kim, Il Hwan Oh.

**Project administration:** Kyueng-Whan Min.

**Resources:** Kyueng-Whan Min, Eun-Kyung Kim, Il Hwan Oh.

**Software:** Kyueng-Whan Min.

**Supervision:** Kyueng-Whan Min.

**Validation:** Kyueng-Whan Min, Kyoung Min Moon.

**Visualization:** Kyueng-Whan Min.

**Writing – original draft:** Kyueng-Whan Min.

**Writing – review & editing:** Kyueng-Whan Min.

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
