## [Decision Letter · Decision Letter 0]

17 Nov 2020

PONE-D-20-32020

High SLC2A1 expression with retinoic acid-induced inhibition promoted cancer survival by suppressing CD8 T cells and B cells in gastric cancer

PLOS ONE

Dear Dr. Son,

Thank you for submitting your manuscript to PLOS ONE. After careful consideration, we feel that it has merit but does not fully meet PLOS ONE’s publication criteria as it currently stands. Therefore, we invite you to submit a revised version of the manuscript that comprehensively addresses the points raised during the review process.

We look forward to receiving your revised manuscript.

Kind regards,

Michael Schubert

Academic Editor

PLOS ONE

3. In the ethics statement in the manuscript and in the online submission form, please provide additional information about the patient records/samples used in your retrospective study, including: a) whether all data were fully anonymized before you accessed them; b) the date range (month and year) during which patients' medical records/samples were accessed.

5. Please provide additional information about each of the cell lines used in this work, including any quality control testing procedures (authentication, characterisation, and mycoplasma testing). For more information, please see http://journals.plos.org/plosone/s/submission-guidelines#loc-cell-lines.

6. Please provide accession numbers and/or URLs for the datasets obtained from the TCGA and GDSC databases.

7. At this time, we ask that you please provide scale bars on the microscopy images presented in Figures 1 and 5, and refer to the scale bar in the corresponding Figure legend.

8. Thank you for stating the following in the Financial Disclosure section: 

 "YES

This study was supported by Daewon Pharmaceutical Co., Ltd. in 2018. (To Byoung Kwan Son) ".

We note that one or more of the authors have an affiliation to the commercial funders of this research study : [Daewon Pharmaceutical Co., Ltd].

Reviewers' comments:

Reviewer's Responses to Questions

**Comments to the Author**

1. Is the manuscript technically sound, and do the data support the conclusions?

Reviewer #1: Partly

Reviewer #2: Partly

2. Has the statistical analysis been performed appropriately and rigorously? 

Reviewer #1: Yes

Reviewer #2: Yes

3. Have the authors made all data underlying the findings in their manuscript fully available?

Reviewer #1: Yes

Reviewer #2: No

4. Is the manuscript presented in an intelligible fashion and written in standard English?

Reviewer #1: Yes

Reviewer #2: Yes

5. Review Comments to the Author

Reviewer #1: • In my opinion the title is not comprehensible, please improve it.

• SLC2A1 is an acronym, and as such the first time it is mentioned it should be made explicit, do the same for all the others throughout the text.

• Retinoic acid, tretinoin: please uniform the names for ATRA.

• What is the purpose of Supplementary table 1? Please include all information for all the patients, also as an average values between groups.

• Mat&met Supplementary information, 1µM retinoic acid: this is too generic definition, please include what kind of retinoic acid is (ATRA? 9-cis? 13-cis?) and the producer.

• Why for RT-PCR GAPDH was used, and instead for immunoblotting beta-actin? Please explain or justify as limitation.

• “MKN-45 cells were seeded at 3,400 cells per well” please add the type of plate.

• Results, “189 normal mucosa, 279 primary cancer and 58 metastatic cancer samples […] primary tumour or metastatic cancer samples” in light of this it is necessary to give some more and correct information about included patients and type of isolated tissue from them, because only 279 were depicted above, if the samples are 279 primary cancer ‘or’ 58 metastatic samples, patients should be more. It should be indicated by how many patients all three types, or only two, or only one samples were taken, justify why the healthy samples are less than the tumors, and so on.

• Table 2: please define all the acronyms throughout the text, for example here CI and HR.

• Figure 1: please define all the elements in the figure in the captions, here for example the numbers of low and high.

• Fig.1b: please add statistic direction

• “In the EHC, high SLC2A1 expression was significantly associated with advanced T stage, advanced N stage, large tumor size, diffuse type, high histological grade, lymphatic invasion, high PD-L1 expression, low PNI, and chemoresistance (p = 0.001, 0.001, 0.003, 0.002, 0.001, 0.001, 0.028, 0.048 and 0.002, respectively) (Table 1).” Please explain among which groups the comparison was made.

• “SLC2A1 expression in relation to mutation, copy number alteration, and methylation status” for this paragraph some things are not explained. For example, which sample group does this analysis belong to? It is not clear why for wild type SLC2A1 expression value is 11, and so on. Add information in mat&met and in captions.

Reviewer #2: The authors study SLC2A1 and possible chemotherapy agents targeting SLC2A1 in gastric cancer.

The manuscript is in general well written and complies with ethical standards of research on human tissue specimen. I feel a few minor changes throughout the manuscript would improve the quality of the manuscript and make it easier understandable for the readers.

Major suggestions:

Study setting:

The headline states that “High SLC2A1 expression with retinoic acid-induced inhibition promoted cancer survival by suppressing CD8 T cells and B cells in gastric cancer”, this is a confusing headline. In the manuscript the authors state that high SLC2A1 expression indicates poor prognosis in gastric cancer patients. Although tretinoin reduced the growth of cancer cell lines with high SLC2A1 expression, it is quite a bold statement to say that this promoted cancer survival, when no survival benefit was, nor could have been, shown in this study setting.

As stated in the abstract, the aim of this study was to analyze the survival data but also genetic changes and immune profiles in gastric cancer patients with high SLC2A1 expression and to provide treatment strategies. The conclusion of the study states that strategies making use of SLC2A1 could contribute to better clinical management/research for patients with gastric cancer. The conclusion should clearly answer the aim of the study, which I feel it didn’t.

Results:

The results on CD8+ cells are confusing. In the EHC, CD8+ cells are elevated in patients with high SLC2A1 expression. However, in TCGA, high SLC2A1 expression was associated with decreased CD8 T cells. This is not discussed in the paper. Why do you think this is?

Minor suggestions:

Abstract:

A short description of methods in the abstract section of the manuscript would be good.

Introduction:

One whole paragraph of the introduction goes over results of previous studies comparing GLUT1 and 18f-FDG-PET-CT. There are plenty of studies on GLUT1 and SLC2A1 already published. Since the current study does not investigate PET imaging, this paragraph could be left out or replaced by a short one meaning statement on PET imaging and glucose metabolism in cancer.

The next to last paragraph of introduction states that TCGA divides gastric cancer into five molecular subtypes, I believe four subtypes have been described.

Methods:

Does the Eulji hospital cohort consist of selected or consecutive patients? Further in the results section the authors mention normal mucosa, primary cancer and metastatis cancer samples. Are the normal mucosa samples from the same patients? If so, why only 189 samples when the cohort was 279 patients? Should be clarified in the methods section of the manuscript.

As neoadjuvant chemotherapy is nowadays standard of care in the treatment of gastric cancer, could this affect the results of this study?

The authors have done a great job in defining all abbreviations in the manuscript. However, I cannot find the definition of GSEA (fourth paragraph of results).

Figures are very nice and well representing the results on their own. In figure 1D, would it be possible to represent time as months instead of days as in figure 1C?

6. PLOS authors have the option to publish the peer review history of their article (what does this mean?). If published, this will include your full peer review and any attached files.

Reviewer #1: No

Reviewer #2: No

---

## [Author Response · Author response to Decision Letter 0]

20 Nov 2020

■ GENERAL COMMENTS TO THE EDITORS AND THE REVEIWERS:

We would like to extend our gratitude to you and the reviewers of the “PLOS One” for taking the time and efforts to review our manuscript. Many of the valuable and constructive points you raised truly inspired the authors. After considering the reviewers’ comments, we revised the manuscript and have indicated the corrections and changes made with yellow highlights in the manuscript.

The revision, based on the review team’s collective input, includes a number of positive changes. Based on your guidance, we:

• Enhanced clarity through general revision of the manuscript 

• Added new descriptions and figures

• Revised the title

We now wish to submit the revised manuscript. The specific revisions and corrections made in response to the reviewers’ comments are as follows:

We have uploaded the marked file.

Answer:

As recommended, we revised manuscript format.

Answer:

Answer: We attached the immunoblotting raw data file. Please find the attached.

3. In the ethics statement in the manuscript and in the online submission form, please provide additional information about the patient records/samples used in your retrospective study, including: a) whether all data were fully anonymized before you accessed them; b) the date range (month and year) during which patients' medical records/samples were accessed.

Answer:

As recommended, we added the sentence in “materials and methods” section as follows:

“The patients' medical records and samples were fully anonymized before we accessed them in September 2018.”

Answer:

As recommended, the ethics statement appear in the Methods section

5. Please provide additional information about each of the cell lines used in this work, including any quality control testing procedures (authentication, characterisation, and mycoplasma testing). For more information, please see http://journals.plos.org/plosone/s/submission-guidelines#loc-cell-lines.

Answer: 

The paper included the purchase information about the cell lines we used. We purchased the cells from the “Korean Cell Line Bank,” a research institute that acquired the status of the International Depository Authority and the institute sent the cells after quality control. For more information on cell lines, please refer to the following sites: 

https://cellbank.snu.ac.kr/main/tmpl/sub_main.php?m_cd=6&m_id=0201&sp=2&c_id=558

6. Please provide accession numbers and/or URLs for the datasets obtained from the TCGA and GDSC databases.

Answer: 

As recommended, we added new URLs in materials and methods as follows:

TCGA data: https://portal.gdc.cancer.gov/

GDSC data: https://www.cancerrxgene.org/celllines

7. At this time, we ask that you please provide scale bars on the microscopy images presented in Figures 1 and 5, and refer to the scale bar in the corresponding Figure legend.

Answer:

As recommended, scar bars were added in figures 1, 3 and 5

8. Thank you for stating the following in the Financial Disclosure section:

 "YES

This study was supported by Daewon Pharmaceutical Co., Ltd. in 2018. (To Byoung Kwan Son) ".

We note that one or more of the authors have an affiliation to the commercial funders of this research study : [Daewon Pharmaceutical Co., Ltd].

Answer:

The funding organization did not play any role in our study and provided only financial support. We have reviewed the author roles in the Author Contribution section and confirm the previous statements are correctly stated.

Answer:

As recommended, we added a new sentence in the “Competing Interests Statement” section as follows.

“This does not alter our adherence to PLOS ONE policies on sharing data and materials.”

Reviewers' comments:

Reviewer's Responses to Questions

Comments to the Author

1. Is the manuscript technically sound, and do the data support the conclusions?

Reviewer #1: Partly

Reviewer #2: Partly

2. Has the statistical analysis been performed appropriately and rigorously?

Reviewer #1: Yes

Reviewer #2: Yes

3. Have the authors made all data underlying the findings in their manuscript fully available?

Reviewer #1: Yes

Reviewer #2: No

4. Is the manuscript presented in an intelligible fashion and written in standard English?

Reviewer #1: Yes

Reviewer #2: Yes

5. Review Comments to the Author

Reviewer #1: 

• In my opinion the title is not comprehensible, please improve it.

Answer:

We revised the title as follows:

High SLC2A1 expression with retinoic acid-induced inhibition promoted cancer survival by suppressing CD8 T cells and B cells in gastric cancer ->

High SLC2A1 expression associated with suppressing CD8 T cells and B cells promoted cancer survival in gastric cancer

• SLC2A1 is an acronym, and as such the first time it is mentioned it should be made explicit, do the same for all the others throughout the text.

Answer:

As recommended, we added the full name of SLC2A1 as “solute carrier family 2 member 1 (SLC2A1)”

• Retinoic acid, tretinoin: please uniform the names for ATRA.

Answer:

As the reviewer well pointed out, we revised as ATRA. 

• What is the purpose of Supplementary table 1? Please include all information for all the patients, also as an average values between groups.

Answer:

To avoid confusion among readers, we removed the supplementary table 1

• Mat&met Supplementary information, 1µM retinoic acid: this is too generic definition, please include what kind of retinoic acid is (ATRA? 9-cis? 13-cis?) and the producer.

Answer: 

Of the many kinds of retinotic acid, we used all-trans-retinoic acid (ATRA). The relevant Information, place of purchase, and product number was inserted in the Materials and methods section as follows: 1 μM retinoic acid (ATRA, all-trans-retinoic acid, R2625, Sigma)

• Why for RT-PCR GAPDH was used, and instead for immunoblotting beta-actin? Please explain or justify as limitation.

Answer: 

Genes such as GAPDH, beta-actin, and alpha-tubulin are all widely used in experiments as housekeeping genes. Unless there are special conditions (such as separating cytosol and nucleus, experiments that can change actin, etc..), it is safe to use any housekeeping gene. As many scholars prefer bata-actin in immunoblotting and GAPDH in PCR (including realtime PCR), we have used it accordingly. I have attached two papers that used actin in immunoblotting and GAPDH in PCR for more information. They also use two housekeeping genes in a paper.

Figure 4 and 5 of J Biol Chem. 2015 Jul 3;290(27):17029-40.

Figure 1D and 1E of PLoS One. 2015; 10(8): e0128943.

• “MKN-45 cells were seeded at 3,400 cells per well” please add the type of plate.

Answer: 

Although the experiment was conducted at 96 well plate, we have not specified the information in the script. Thanks to the sharp point by the reviewer, we added the information as follows: MKN-45 cells were seeded at 3,400 cells per well in 96 well plate.

• Results, “189 normal mucosa, 279 primary cancer and 58 metastatic cancer samples […] primary tumour or metastatic cancer samples” in light of this it is necessary to give some more and correct information about included patients and type of isolated tissue from them, because only 279 were depicted above, if the samples are 279 primary cancer ‘or’ 58 metastatic samples, patients should be more. It should be indicated by how many patients all three types, or only two, or only one samples were taken, justify why the healthy samples are less than the tumors, and so on.

Answer:

Metastatic cancer, primary cancer, and normal tissue were collected from one patient. In the process of making tissue microarray, there were many cases of missing normal tissues.

We revised the original sentence in the “Clinical manifestations of SLC2A1” section as follows:

“We have analyzed 189 normal and 58 metastatic tumor samples from a total 279 primary cancer samples.”

• Table 2: please define all the acronyms throughout the text, for example here CI and HR.

Answer:

We added the explanation as “HR, hazard ratio; CI, confidence interval” in the table 2. 

• Figure 1: please define all the elements in the figure in the captions, here for example the numbers of low and high.

Answer:

We appreciate your insight. 

We added “low <0.5” and “high >0.5” in EHC.

We added “low <10.2” and “high >10.2” in TCGA.

• Fig.1b: please add statistic direction

Answer:

We added the p value in the Fig 1b.

• “In the EHC, high SLC2A1 expression was significantly associated with advanced T stage, advanced N stage, large tumor size, diffuse type, high histological grade, lymphatic invasion, high PD-L1 expression, low PNI, and chemoresistance (p = 0.001, 0.001, 0.003, 0.002, 0.001, 0.001, 0.028, 0.048 and 0.002, respectively) (Table 1).” Please explain among which groups the comparison was made.

Answer:

We added “compared with low SLC2A1 expression” in the sentence.

• “SLC2A1 expression in relation to mutation, copy number alteration, and methylation status” for this paragraph some things are not explained. For example, which sample group does this analysis belong to? It is not clear why for wild type SLC2A1 expression value is 11, and so on. Add information in mat&met and in captions.

Answer:

This is an analysis based on the TCGA data for SLC2A1. It means that the mRNA value of SLC2A1 for wild type is 11.

Reviewer #2: The authors study SLC2A1 and possible chemotherapy agents targeting SLC2A1 in gastric cancer.

The manuscript is in general well written and complies with ethical standards of research on human tissue specimen. I feel a few minor changes throughout the manuscript would improve the quality of the manuscript and make it easier understandable for the readers.

Major suggestions:

Study setting:

The headline states that “High SLC2A1 expression with retinoic acid-induced inhibition promoted cancer survival by suppressing CD8 T cells and B cells in gastric cancer”, this is a confusing headline. In the manuscript the authors state that high SLC2A1 expression indicates poor prognosis in gastric cancer patients. Although tretinoin reduced the growth of cancer cell lines with high SLC2A1 expression, it is quite a bold statement to say that this promoted cancer survival, when no survival benefit was, nor could have been, shown in this study setting.

Answer:

We revised the title as follows:

High SLC2A1 expression with retinoic acid-induced inhibition promoted cancer survival by suppressing CD8 T cells and B cells in gastric cancer ->

High SLC2A1 expression associated with suppressing CD8 T cells and B cells promoted cancer survival in gastric cancer

As stated in the abstract, the aim of this study was to analyze the survival data but also genetic changes and immune profiles in gastric cancer patients with high SLC2A1 expression and to provide treatment strategies. The conclusion of the study states that strategies making use of SLC2A1 could contribute to better clinical management/research for patients with gastric cancer. The conclusion should clearly answer the aim of the study, which I feel it didn’t.

Answer:

We revised the conclusion as follows:

“Treatment involving the use of SLC2A1 could contribute to better clinical management/research for patients with gastric cancer.”

We revised the aim as follows: (treatment strategies -> treatment)

The aim of this study was to analyze the survival and genetic changes/immune profiles in patients with gastric cancer with high SLC2A1 expression and to provide treatment for improving prognosis.

Results:

The results on CD8+ cells are confusing. In the EHC, CD8+ cells are elevated in patients with high SLC2A1 expression. However, in TCGA, high SLC2A1 expression was associated with decreased CD8 T cells. This is not discussed in the paper. Why do you think this is?

Answer:

In the two EHC and TCGA cohorts, the high SLC2A1 expression group had a decrease in CD8+ T cells.

Minor suggestions:

Abstract:

A short description of methods in the abstract section of the manuscript would be good.

Introduction:

One whole paragraph of the introduction goes over results of previous studies comparing GLUT1 and 18f-FDG-PET-CT. There are plenty of studies on GLUT1 and SLC2A1 already published. Since the current study does not investigate PET imaging, this paragraph could be left out or replaced by a short one meaning statement on PET imaging and glucose metabolism in cancer.

Answer:

As recommended, we removed the contents.

The next to last paragraph of introduction states that TCGA divides gastric cancer into five molecular subtypes, I believe four subtypes have been described.

Answer:

As you pointed out, we revised as “four”.

Methods:

Does the Eulji hospital cohort consist of selected or consecutive patients? Further in the results section the authors mention normal mucosa, primary cancer and metastatis cancer samples. Are the normal mucosa samples from the same patients? If so, why only 189 samples when the cohort was 279 patients? Should be clarified in the methods section of the manuscript.

Answer:

Metastatic cancer, primary cancer, and normal tissue were collected from one patient. In the process of making tissue microarray, there were many cases of missing normal tissue.

We revised the sentence in the “Clinical manifestations of SLC2A1” section as follows:

“We have analyzed 189 normal and 58 metastatic tumor samples from a total 279 primary cancer samples.”

As neoadjuvant chemotherapy is nowadays standard of care in the treatment of gastric cancer, could this affect the results of this study?

Answer: 

no

We focused mainly on stomach cancer with high SLC2A1 expression.

The authors have done a great job in defining all abbreviations in the manuscript. However, I cannot find the definition of GSEA (fourth paragraph of results).

Answer:

The definition is provided in the 1st paragraph in the materials and method section. 

-> gene set enrichment analysis (GSEA version 4.3)

Figures are very nice and well representing the results on their own. In figure 1D, would it be possible to represent time as months instead of days as in figure 1C?

Answer:

We revised as months in the Fig 1D.

6. PLOS authors have the option to publish the peer review history of their article (what does this mean?). If published, this will include your full peer review and any attached files.

Do you want your identity to be public for this peer review? For information about this choice, including consent withdrawal, please see our Privacy Policy.

Reviewer #1: No

Reviewer #2: No

---

## [Decision Letter · Decision Letter 1]

22 Dec 2020

High SLC2A1 expression associated with suppressing CD8 T cells and B cells promoted cancer survival in gastric cancer.

PONE-D-20-32020R1

Dear Dr. Son,

We’re pleased to inform you that your manuscript has been judged scientifically suitable for publication and will be formally accepted for publication once it meets all outstanding technical requirements.

Kind regards,

Michael Schubert

Academic Editor

PLOS ONE

Reviewers' comments:

Reviewer's Responses to Questions

**Comments to the Author**

1. If the authors have adequately addressed your comments raised in a previous round of review and you feel that this manuscript is now acceptable for publication, you may indicate that here to bypass the “Comments to the Author” section, enter your conflict of interest statement in the “Confidential to Editor” section, and submit your "Accept" recommendation.

Reviewer #1: All comments have been addressed

Reviewer #2: All comments have been addressed

2. Is the manuscript technically sound, and do the data support the conclusions?

Reviewer #1: Yes

Reviewer #2: Partly

3. Has the statistical analysis been performed appropriately and rigorously? 

Reviewer #1: Yes

Reviewer #2: Yes

4. Have the authors made all data underlying the findings in their manuscript fully available?

Reviewer #1: Yes

Reviewer #2: Yes

5. Is the manuscript presented in an intelligible fashion and written in standard English?

Reviewer #1: Yes

Reviewer #2: Yes

6. Review Comments to the Author

Reviewer #1: (No Response)

Reviewer #2: The authors have answered my questions and addressed the few concerns regarding their manuscript in a satisfactory manner.

7. PLOS authors have the option to publish the peer review history of their article (what does this mean?). If published, this will include your full peer review and any attached files.

Reviewer #1: No

Reviewer #2: No

---

## [Editor Report · Acceptance letter]

18 Jan 2021

PONE-D-20-32020R1 

High SLC2A1 expression associated with suppressing CD8 T cells and B cells promoted cancer survival in gastric cancer. 

Dear Dr. Son:

I'm pleased to inform you that your manuscript has been deemed suitable for publication in PLOS ONE. Congratulations! Your manuscript is now with our production department. 

Kind regards, 

on behalf of

Dr. Michael Schubert 

Academic Editor

PLOS ONE